# The Association between Breastmilk Glucocorticoid Concentrations and Macronutrient Contents Throughout the Day

**DOI:** 10.3390/nu11020259

**Published:** 2019-01-24

**Authors:** Jonneke J. Hollanders, Stefanie M. P. Kouwenhoven, Bibian van der Voorn, Johannes B. van Goudoever, Joost Rotteveel, Martijn J. J. Finken

**Affiliations:** 1Room ZH 9 D 36, Department of Pediatrics, Emma Children’s Hospital, Amsterdam UMC, Vrije Universiteit Amsterdam, P.O. Box 7057, 1007 MB Amsterdam, The Netherlands; s.kouwenhoven@vumc.nl (S.M.P.K.); b.vandervoorn@erasmusmc.nl (B.v.d.V.); h.vangoudoever@vumc.nl (J.B.v.G.); j.rotteveel@vumc.nl (J.R.); m.finken@vumc.nl (M.J.J.F.); 2Department of Pediatric Endocrinology, Sophia Children’s Hospital, Erasmus MC, P.O. Box 2060, 3000 CA Rotterdam, The Netherlands

**Keywords:** glucocorticoids, cortisol, cortisone, macronutrients, human milk, diurnal rhythm, circadian rhythm, fat, carbohydrates, protein

## Abstract

Background: Glucocorticoids (GCs) in breastmilk follow the maternal hypothalamus–pituitary–adrenal axis activity and may affect the offspring’s growth and neurodevelopment. There is some evidence suggesting that macronutrients in breastmilk also fluctuate throughout the day. We aimed to research whether GCs and macronutrients are correlated in multiple breastmilk samples obtained over a 24-h period. Methods: A total of 10 mothers provided 45 breastmilk samples collected over a 24-h period. Cortisol and cortisone levels were determined by LC–MS/MS, and macronutrients were measured with mid-infrared spectroscopy. Correlations between breastmilk GCs and macronutrients were assessed with Pearson correlations and linear mixed models. Results: No associations were found between breastmilk GCs and macronutrients (cortisol: β-0.1 (95% confidence interval: −1.0 to 0.7), −4.9 (−12.9 to 3.1) for fat, protein, and carbohydrates, respectively; and −0.3 (−5.6 to 5.0) and cortisone: 0.0 (−2.5 to 2.5), −17.4 (−39.8 to 5.0), and −2.7 (−17.7 to 12.3)) for fat, protein, and carbohydrates, respectively. Adjusting for the time of collection to account for GC rhythmicity did not change the results. Conclusion: We found no associations between GCs and macronutrients in human breastmilk. The excretion of GCs in breastmilk and the effects of breastmilk GCs on offspring are, therefore, likely independent of the excretion and effects of the macronutrients.

## 1. Introduction

It has been well established that breastmilk is the preferred nutrition for neonates. The composition of breastmilk shows a wide variability between mothers [1,2]. Moreover, the nutritional composition of milk changes throughout the lactation period [2,3], during the day [4,5] and within the same feed [3,5], to meet the requirements of the infant. In short, protein concentrations decrease throughout the lactation period, while lactose and fat remain mostly stable, although some variation in fat concentrations has been observed [2]. Over the day, fat concentrations show a diurnal rhythm, with higher concentrations during the day compared with the morning, evening, and night. In contrast, protein and lactose concentrations remain stable throughout the day. Additionally, during a feed, fat concentrations increase depending on the degree of breast emptying [6], while no effects on lactose or protein [5,7] have been observed. The excretion of milk macronutrients is a complex and active process, including transcellular transport, as well as intracellular synthesis [8].

The beneficial effects of breastmilk result not only from personalized, nutritive composition, but also from non-nutritive bioactive factors [9], such as glucocorticoids (GCs). In breastmilk, unlike serum, cortisone is much more abundant than cortisol, probably due to the expression of 11β-HSD type 2 in the mammary gland, analogous to the salivary gland [10]. Breastmilk GCs have been associated with growth and neurodevelopment in both animal and human studies (as reviewed in [11]). However, the results were contradictory: both a more and less confident temperament was reported with higher cortisol levels in rhesus macaques, while in rats, better stress resilience and less fearfulness was found. In man, cortisol has been associated with autonomic stability, as measured with the Neonatal Behavioral Assessment Scale, and with the negative affectivity dimension of the Infant Behavior Questionnaire. Additionally, in rhesus macaques, cortisol has been associated with weight gain in offspring, while in man, an inverse correlation with body mass index (BMI) has been found.

Studies of the breastmilk of rhesus monkeys have found a positive correlation between GC levels and milk protein, as well as fat concentrations [12,13]. Additionally, although GCs are lipophilic, 70–85% of the GCs measured in a single sample of cow’s milk were shown to be associated with the skim fraction of the milk [14]. Although associations with nutritional content have not yet been studied in humans, these findings are confirmed by our liquid chromatography–tandem mass spectrometry (LC–MS/MS) method for breastmilk GC analyses, which showed that the removal of undesired lipids by hexane washings did not influence the milk’s GC content [15]. This might be due to the binding of GCs to proteins, such as corticosteroid-binding globulin (CBG), CBG-like proteins [16], and albumin [17]. The excretion of GCs in milk is currently not fully understood, although they are possibly subject to passive diffusion, owing to their lipophilic structure [11].

We have recently shown that the GCs cortisol and cortisone in breastmilk show a diurnal rhythm, which is correlated to the maternal hypothalamic–pituitary–adrenal (HPA) axis activity [18], with high concentrations in the early morning, which decline to a nadir at night. These findings were replicated by another group [19]. It is currently unknown whether this diurnal rhythm exerts any effects on the offspring. Additionally, it is not known whether this rhythm is correlated with milk macronutrients. If milk GCs appear to be correlated with macronutrients, their effects on the offspring may be (partly or wholly) explained by the nutritional composition of breastmilk. We therefore measured GCs, as well as macronutrients, in samples obtained before every feed over a 24-h period to determine whether GC concentrations are correlated with macronutrients. We hypothesized that GC concentrations, due to protein binding, are correlated with milk protein concentrations but not with other macronutrients.

## 2. Methods

### 2.1. Study Population

The subjects in this study were a subset of mothers who participated in the Cortisol in Mother’s Milk (CosMos) study, which included mother–infant pairs between March 2016 and July 2017 from both the Onze Lieve Vrouwe Gasthuis (OLVG) and the Amsterdam University Medical Center, location VU University Medical Center (VUmc)in Amsterdam. The primary aim of the study was to research the effect of breastmilk GC rhythmicity on infant HPA axis activity, behavior, and body composition. Women included at the OLVG were monitored at the Psychiatric Obstetric Pediatric (POP) outpatient clinic due to previous or current psychopathologic complaints. We included these mothers to obtain a diverse population, because psychopathology is often accompanied by changes in HPA axis activity [20,21]. Mother–infant pairs were eligible for inclusion if the infant was born at term age (37–42 weeks) with a normal birth weight (−2 to +2 SD score) and if the mother had the intention of exclusively breastfeeding the infant for ≥3 months. Subjects were excluded from the study for the following reasons: major congenital malformations, multiple pregnancy, pre-eclampsia, maternal alcohol consumption of >7 IU (International Unit) IU/wk, and/or a fever (temperature > 38.5 °C) at 1 month postpartum. Medication use other than “over the counter” drugs was also an exclusion criterion, except for anti-depressant use in the mothers included at the OLVG. The approval of the Medical Ethics Committee of the VUmc was obtained (protocol number 2015.524), and written informed consent was obtained from all the participating mothers.

### 2.2. Milk Sampling

At 1 month postpartum (±5 days), the mothers were asked to collect a milk sample before every feed during a 24-h period. The milk was collected using a breast pump or via manual expression. The mothers were asked to collect 1–2 mL of milk. Milk samples were eligible for macronutrient analysis when the mothers collected >10 mL of milk for several or all of the samplings. This resulted in 45 random samples, which were used for this study. 

The milk was stored in the mother’s freezer until transportation to the laboratory, where it was stored at −20 °C. The samples were thawed twice—once for the GC analyses and once more for the macronutrient analyses—and were stored at −20 °C in between analyses for several months.

At the same time, the participants were asked to fill in the Hospital Anxiety and Depression Scale (HADS), which is an index to measure clinically relevant anxiety and/or depression symptoms in patients from non-psychiatric hospitals [22]. A score of ≥8 on the depression (HDS) or anxiety (HAS) subscale is indicative of an elevated psychological stress level.

### 2.3. Laboratory

#### 2.3.1. GC Concentrations

The total cortisol and cortisone concentrations in the breastmilk were determined by isotope dilution liquid chromatography–tandem mass spectrometry (LC–MS/MS) as previously published [15]. In short, the breastmilk samples were washed 3 times with 2 mL of hexane to remove lipids after adding internal standards (13C3-labeled cortisol and 13C3-labeled cortisone). Then, samples were extracted and analyzed by XLC-MS/MS13, a Symbiosis online SPE system (Spark Holland, Emmen, The Netherlands) coupled to a Quattro Premier XE tandem mass spectrometer (Waters Corp., Milford, MA, USA). The intra-assay coefficients of variation (CV%) were 4 and 5% for cortisol levels of 7 and 23 nmol/L and 5% for cortisone levels of 8 and 33 nmol/L for the LC-MS/MS measurements, while the inter-assay CV% was <9% and the lower limit of quantitation was 0.5 nmol/L for both cortisol and cortisone.

#### 2.3.2. Macronutrient Analysis

First, the milk samples were homogenized with an ultrasonic processor (VCX130, Sonics & Materials Inc, Newtown, CT, USA; 98% amplitude for 7.0 s) after heating the sample to 40 °C. Next, the fat, protein, and carbohydrate concentrations (in g/100 mL), as well as the total solids and energy, were analyzed simultaneously with a human milk analyzer (MIRIS, Uppsala, Sweden) through the use of mid-infrared spectroscopy. After 10 samples, the milk analyzer was recalibrated. The milk samples were analyzed in duplicate or, when possible, triplicate (*n* = 4) and were subsequently averaged. The protein concentrations were expressed as crude or true protein. Crude protein is based on the total amount of nitrogen in the sample, which also includes non-protein nitrogen compounds (20–25% of the total nitrogen in human milk). The true protein levels were calculated by the software as 80% of the crude protein concentrations. For our analyses, the true protein concentrations were used. The energy content (kcal/100 mL) was calculated with the formula: 9.25 × fat + 4.40 × crude protein + 3.95 × carbohydrate. 

### 2.4. Statistics

First, GC, fat, carbohydrate, and protein concentrations were plotted for subjects with >5 samples to visualize the data (*n* = 4).

Next, the correlations between the milk macronutrients and milk GCs were analyzed with Pearson correlations. Additionally, to account for the repeated measurements within each subject, linear mixed models were performed with cortisol or cortisone concentration as the dependent variable and the macronutrient concentration (fat, protein, or carbohydrates) or total energy as the fixed effect, with the subject added as a random effect. The analyses were repeated while adjusting for the time of collection as a fixed effect.

## 3. Results

### 3.1. Study Population

A random subset of 10 mothers collected a total of 45 samples with >10 mL of milk; the sampling took place between 26 and 36 days postpartum. None of the mothers used antidepressants, and only one mother had an elevated HAS score (11 points). The mothers gave birth to 5 boys and 5 girls, with a gestational age between 37 + 1 and 41 + 2 weeks and a birth weight between 2726 and 4030 g. A total of 5 mothers delivered their babies via caesarian section. 

### 3.2. Individual Plots

Figure 1 shows the individual plots for 4 mothers, who had >5 samples of >10 mL of breastmilk. While in all the mothers a diurnal rhythm of cortisol and cortisone can be seen, no rhythm appeared to be present for fat, carbohydrates, and protein. 

### 3.3. Correlations

The cortisol and cortisone concentrations were highly correlated (Table 1; *p* < 0.001), while the GC concentrations were not correlated with the fat, protein, or carbohydrate contents of the milk or with the total solids or milk energy.

The fat concentrations, carbohydrate concentrations, total solids, and energy were highly correlated. The protein levels were correlated with fat concentrations, total solids, and energy but not with the carbohydrate concentrations.

### 3.4. Linear Mixed Models

No significant associations were found between the milk GC concentrations and the milk macronutrients, total solids, or milk energy (Table 2). Adjusting for the time of collection did not alter these results.

Using crude protein concentrations instead of true protein levels did not change the results (data not shown).

## 4. Discussion

To the best of our knowledge, this study is the first to assess the correlations between the diurnal pattern of cortisol and cortisone in breastmilk and the milk macronutrients fat, protein and carbohydrates, as well as energy in humans. We did not find any correlations between breastmilk GCs and macronutrients, even after taking the diurnal rhythm of breastmilk GCs into account. The excretion of GCs in breastmilk and the effects of GCs on offspring are, therefore, likely to be independent of macronutrients.

Previously, studies in rhesus macaques showed correlations between milk glucocorticoids and both the fat and protein contents of milk [12,13]. This discrepancy with our study might be due to several reasons. First, both Sullivan et al. [13] and Hinde et al. [12] emptied the mammary gland entirely, while for ethical reasons, the mothers in our study collected only a small portion of milk before feeding their infants. Although previous research has shown that GC concentrations are similar in fore- and hindmilk [23] and that lactose and protein levels also do not change during a feeding [5,7], fat levels do increase during a feed [6]. Additionally, both studies only took one sample of milk per subject, and only Hinde et al., (2015) [12] minimized the time window of the sample collection (between 11:30–13:00 after 3.5–4 h of milk accumulation). This sampling period most likely did not measure the peak cortisol levels, because cortisol concentrations in rhesus macaques reach their maximum levels around 8:00 [24], and the GC concentrations are, therefore, likely to be less variable in those studies as compared with ours. However, when we reanalyzed our data with only the samples collected after 11:30, we still found no association between milk GCs and macronutrients [data not shown]. Schwalm et al. (1978) [14] researched cow’s milk and concluded that no positive correlation was found between milk glucocorticoids and fat, while there was also no correlation found with milk protein. No previous study has researched the correlations between milk GCs and macronutrients in human milk. 

We did not find a correlation between GCs and milk protein, as hypothesized. Although 70–85% of GCs in cow’s milk are associated with the skim fraction of the milk [14], the concentration of GCs in milk is low. Moreover, only a small proportion of the protein fraction in milk harbors CBG, CBG-like proteins, or albumin [16,25]. It is, therefore, still possible that GCs are correlated with specific proteins but not with the total protein concentration. Nevertheless, CBG-binding activity in milk was also possibly found to exhibit a diurnal rhythm [26], with a peak in the evening, which did not coincide with the GC peak. This, however, should not have hampered our GC measurements, as our laboratory method assesses total GC concentrations [11]. Lastly, it is unlikely that GCs could be correlated with fat concentrations, because the removal of undesired lipids by hexane washings during our LC–MS/MS method did not influence the breastmilk’s GC content [15], although the free fraction of breastmilk GCs could hypothetically be correlated with the lipid fraction.

Our findings could have several implications. First, it appears that the excretion of GCs in milk is not dependent on the macronutrients and vice versa. Indeed, due to the lipophilic structure of GCs, they are probably subject to passive diffusion from the systemic circulation into the breastmilk. This assumption was corroborated by our previous research, showing that GCs in breastmilk closely follow the maternal HPA axis activity [18]. In contrast, the secretion of macronutrients into breastmilk is an active and complex process, with both transcellular transport and intracellular synthesis. Proteins and lactose are transported in secretory vehicles and secreted via exocytosis, while lipids are released in milk fat globules [8]. The lack of association between breastmilk macronutrients and GCs may suggest that the latter do not influence these transport mechanisms, at least not acutely. Second, several studies have been published recently suggesting that the GCs in breastmilk affect neurodevelopment and growth in humans [27,28,29]. Because milk GCs and macronutrients are not associated, it is unlikely that these effects could have been falsely attributed to GCs. However, the underlying mechanism is not yet understood fully but might be due to a direct influence of breastmilk GCs on the infant’s HPA axis activity and development or through an interaction with the gut microbiome (i.e., the “gut–brain axis hypothesis”) [11,30,31]. Therefore, further research to assess these associations is still warranted, especially because these studies did not take the diurnal rhythm of GCs in breastmilk into account.

Our study has several strengths and limitations. First, this is the first study to research the correlation between GCs and macronutrients in human milk. Additionally, the diurnal rhythm of GCs in breastmilk was taken into account by sampling multiple times over the day, allowing for a more precise analysis of the correlation between breastmilk GCs and macronutrients. However, our study also had limitations. Because mothers were asked to collect milk prior to each feeding, only foremilk was collected. Preferably, a portion of a completely pumped feed would be analyzed, which would also include the hindmilk. Next, the CosMos study aimed to research the association between milk GCs and effects on offspring. The current study used leftover samples from mothers who collected >10 mL of milk rather than the requested 1–2 mL. The sample size of this study was therefore limited. Moreover, the samples were thawed twice—once for the GC analyses and once more for the macronutrient analyses—and were stored at −20 °C in between analyses for several months. While GCs in milk are highly stable [15], macronutrients are probably more vulnerable to degradation, with studies showing a decrease in fat concentrations of up to 10%, while other macronutrients remained stable or showed minimal degradation (up to 2%); these effects were already observed after 2–7 days of freezing [32,33,34,35,36,37]. It is therefore possible that the macronutrient concentrations were initially different than what was measured during the analyses. However, this would lead to a systematic error rather than a random error, and it is therefore likely that the correlations would still be absent even if the macronutrient analyses had been performed with fresh breastmilk. The results of our study should, therefore, be interpreted with caution due to all of these factors.

In conclusion, no associations between GCs and macronutrients in human milk were found in this study. The excretion of GCs in breastmilk and the effects of breastmilk GCs on offspring are, therefore, likely independent of the excretion and effects of the macronutrients.

## Figures and Tables

**Figure 1 nutrients-11-00259-f001:**
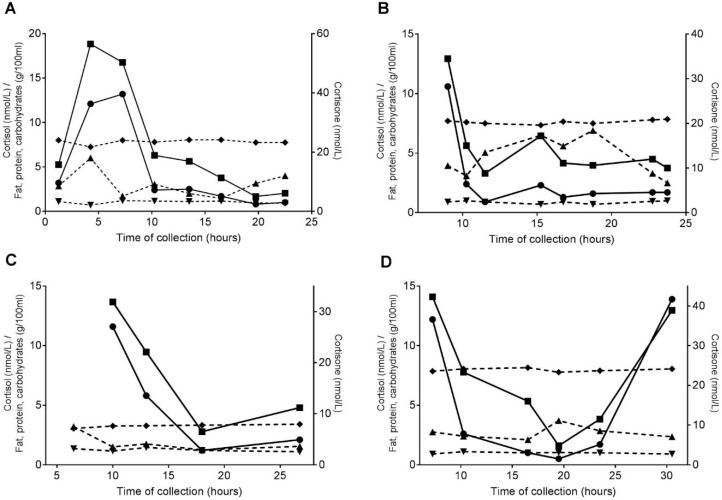
(**A**–**D**) The individual plots for milk cortisol (●), cortisone (⬛), fat (▲), protein (▼), and carbohydrate (◆) concentrations for four mothers. Left y-axis: cortisol concentrations (nmol/L) and milk macronutrient concentrations (g/100 mL); right y-axis: cortisone concentrations (nmol/L); x-axis: time of collection (hours); continuous lines: GC concentrations; and dotted lines: macronutrient concentrations.

**Table 1 nutrients-11-00259-t001:** Correlations between the milk glucocorticoids and macronutrients.

	Cortisol	Cortisone	Fat	Protein	Carbohydrates	Total Solids	Energy
Cortisol	1						
Cortisone	0.827 *	1					
Fat	−0.096	−0.050	1				
Protein	−0.071	−0.119	−0.455 *	1			
Carbohydrates	−0.016	−0.050	−0.639 *	−0.090	1		
Total solids	−0.127	−0.088	−0.988 *	−0.404 *	−0.564 *	1	
Energy	−0.109	−0.068	−0.998 *	−0.432 *	−0.615 *	0.995 *	1

Values represent Pearson’s r *: *p* value < 0.001.

**Table 2 nutrients-11-00259-t002:** Associations between milk glucocorticoids and macronutrients.

	Cortisol				Cortisone			
	Crude Analyses		Adjusted for Time of Collection		Crude Analyses		Adjusted for Time of Collection	
	β (95% CI)	*p* Value	β (95% CI)	*p* Value	β (95% CI)	*p* Value	β (95% CI)	*p* Value
**Fat**	−0.3 (−1.3 to 0.7)	0.54	−0.1 (−1.0 to 0.7)	0.75	−0.4 (−3.2 to 2.3)	0.75	0.0 (−2.5 to 2.5)	0.998
**Protein**	−2.0 (−11.0 to 6.9)	0.65	−4.9 (−12.9 to 3.1)	0.22	−9.6 (−34.6 to 15.4)	0.44	−17.4 (−39.8 to 5.0)	0.13
**Carbohydrates**	−0.3 (−6.3 to 5.7)	0.92	−0.3 (−5.6 to 5.0)	0.90	−2.7 (−19.4 to 14.1)	0.75	−2.7 (−17.7 to 12.3)	0.72
**Total solids**	−0.5 (−1.7 to 0.7)	0.41	−0.3 (−1.4 to 0.8)	0.55	−1.0 −4.3 to 2.4)	0.57	−0.5 (−3.6 to 2.6)	0.74
**Energy**	0.0 (−0.2 to 0.1)	0.48	0.0 (−0.1 to 0.1)	0.66	−0.1 (−0.4 to 0.3)	0.66	0.0 (−0.3 to 0.3)	0.88

Values represents β (95% CI), analyzed with linear mixed models.

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
