# Peer review of "The Association between Breastmilk Glucocorticoid Concentrations and Macronutrient Contents Throughout the Day"

_nutrients, 2019, doi:10.3390/nu11020259_

Reviewer 1 Report

This manuscript from Hollanders et al examines the hypothesis that glucocorticoid concentrations in human milk may be correlated with milk protein concentration due to binding to CBG or CBG-like proteins. The authors measured cortisol and cortisone and macronutrients in breast milk samples and examined correlations between breast milk GCs and macronutrients. They found no associations between GCs and macronutrients in human milk and conclude that the excretion of GCs in breastmilk, as well as effects of breastmilk GCs, are likely independent of the excretion and effects of macronutrients.

The hypothesis examined in the paper is interesting given the importance of and growing interest in the role of the non-nutritive components of breast milk in human infant growth and development. However, the several issues need to be addressed/discussed:

The authors have hypothesized that GCs – through binding with CBG and CBG-like proteins – may be correlated with milk protein concentrations. They have however not examined expression of these proteins in the samples nor discussed the expression and potential diurnal variation of CBG/CBG-like proteins in human milk. What percentage of the total protein and globulin do these proteins comprise? Although total protein concentration does not appear to vary significantly diurnally, might the concentration of these fractions vary with the time of day?

The authors should discuss the potential binding of glucocorticoids in breast milk to lipids or other macronutrients.  

The authors should also cite the work of other groups (e.g. Pundir et al, 2017) and compare results with their studies.  

The conclusions would be greatly strengthened by the simultaneous examination of GCs and macronutrients in milk samples (which could be divided to perform both measurements and therefore subjected to the same length of freezing and number of freeze-thaws). The differences in storage (as mentioned in the discussion by the authors) may significantly alter protein composition and change results.  

Author Response

Please find our response in the attachments

Reviewer 2 Report

Authors investigated the relationships between concentrations of glucocorticoids (GCs) and macronutrients in human breastmilk. They suggested no associations between GCs and macronutrients in human milk, while the previous studies using rhesus monkeys showed the positive results.

 Comments:

1.     Reviewer could not find the description of informed consent and approval of ethics committee in this MS.

2.     Reviewer could not understand the significance of this study. How GCs in the mother’s milk influence on the children? Is there any data that GCs in the milk influence children’s HPA-axis?

3.     This MS was a negative study, however, even if the diurnal GC rhythm GC in the milk is influenced by macronutrients, the concentrations of GCs may be too low to have some effects on the offspring.

4.     As Authors indicated in the Discussion (line 201), macronutrient concentrations might be unstable during freeze and saw procedures and also long-term storage at – 20 degree. Authors should investigate the stabilities of macronutrients in the milk under these difficult conditions.

5.     Cortisol concentration in the breast milk seemed lower than that of cortisone (Figure 1). It is known that serum cortisol levels are almost 3 times higher than those of cortisone. These differences between serum and milk should be explained. Levels of corticosteroid binding globulin or free fractions of cortisol and cortisone in the breast milk should be measured.

Author Response

(The authors gave the same response as above.)

Reviewer 3 Report

In the current study the authors measured glucocorticoids levels, as well as macronutrients, in 45 milk samples obtained before every feed over a 24-hour period from 10 women.

They aimed to evaluate whether GCs concentrations during the day correlate with macronutrients and in particular with fat, protein and carbohydrates.

They didn’t find any correlations between breastmilk GCs and macronutrients even after time of collection, to account for GC rhythmicity. They concluded that GCs in the milk during the day are independent of the excretion and effects of the macronutrients.

The manuscript is interesting and original, however it is not completely clear the rationale of their study. Why we should expect to find correlations between GCs content and macronutrient? and how their possible correlations may have an impact on offspring?

The main issue of this manuscript is related to the sample size and also to the fact that the analyses are restricted to control women, so to women with no presence of stress related psychiatric disorders.  However, an important missing information is related to the clinical assessment; women have not been clinically evaluated and this is important especially in term of GCs content. A clinical questionnaire to check for mood and for exposure to stressful experiences would be important.

They tested the correlations between GCs and macronutrients in 10 women, 5 of them delivered their baby with a caesarian section. Why did the authors decided to have half women with a caesarian section and the other half with a physiological delivery? and why did they selected 5 women that had boys and 5 women that had girls?

In the text they also mention that the correlation between GCs content with macronutrient maybe important for offspring development; however they don’t check any outcomes in offspring. Could they measure cortisol content in offspring saliva samples?

It will be also interesting to see correlations between GCs and micronutrients, such as folic acid, and other vitamins that are known to be important for the development of the baby.

in Figure 1 they reported individual plots for 4 mothers; however, in the legend is not clear the meaning of the dot lines and the continuous lines. Also, from the figures it seems that there is a representation of concentrations during the day of 5 subjects and not 4; could the authors clarify this?

Author Response

Please find our response in the attachments

Round  2

Reviewer 1 Report

Thanks to the authors for answering the concerns raised. 

Reviewer 2 Report

Reviewer understand the Authors' responses.